

# The operational eEMEP model for volcanic SO₂ and ash forecasting

Birthe M. Steensen, Michael Schulz, Peter Wind, Álvaro M. Valdebenito, Hilde Fagerli

Research department, Norwegian Meteorological Institute, Postbox 43 Blindern, 0313 Oslo, Norway

*Correspondence to*: Birthe M. Steensen (birthe.steensen@met.no)

5 **Abstract.**

This paper presents a new version of the EMEP MSC-W model called eEMEP developed for transportation and dispersion of volcanic emissions, both gases and ash. EMEP MSC-W is usually applied to study problems with air pollution and aerosol transport and requires some adaptation to treat volcanic eruption sources and effluent dispersion. The operational setup of model simulations in case of a volcanic eruption is described. Important choices have to be made to achieve CPU efficiency so that emergency situations can be tackled in time, answering relevant questions of ash advisory authorities. An efficient model needs to balance complexity of the model and resolution. We have investigated here a meteorological uncertainty component of the volcanic cloud forecast by using a consistent ensemble meteorological dataset (GLAMEPS forecast) in three resolutions for the case of SO₂ effusion from the 2014 Barðarbunga eruption. The low resolution (40x40km) ensemble members show larger agreement in plume position and intensity, suggesting that the ensemble here don't give much added value. For comparing the dispersion in different resolutions we compute the area where the column load of the volcanic tracer, here SO₂, is above a certain threshold, varied for testing purposed between 0.25-50 DU Dobson units. The increased numerical diffusion causes a larger area (+34%) to be covered by the volcanic tracer in the low resolution simulations than in the high resolution ones. The higher resolution (10x10km) ensemble members show higher concentrations farther away from the volcanic eruption site in more narrow plumes. Plume positions are more varied between the high resolution members, while the plume form resemble the observed plumes more than the low resolution ones. For a volcanic emergency case this means: To obtain quickly results of the transport of volcanic emissions an individual simulation with our low resolution is sufficient, however, to forecast peak concentrations with more certainty for forecast or scientific analysis purposes a finer resolution is needed. The model is further developed to simulate ash from highly explosive eruptions. A possibility to increase the number of vertical layers, achieving finer vertical resolution, as well as a higher model top is included in the eEMEP version. Ash size distributions may be altered for different volcanic eruptions and assumptions. Since ash particles are larger than typical particles in the standard model, gravitational settling across all vertical layers is included. We attempt finally a specific validation of the simulation of ash and its vertical distribution. Model simulations with and without gravitational settling for the 2010 Eyjafjallajökull eruption are compared to lidar observations over Central Europe. The results show that with gravitation the centre of ash mass can be 1km lower over central Europe than without gravitation. However the height variations in the ash layer caused by real weather situations are not captured perfectly well by either of the two simulations, playing down the role of gravitation parameterization imperfections. Both model simulations have on average ash centre of mass below the observed values. Correlation between the observed and corresponding model centre of



mass are higher for the model simulation with gravitational settling for four of six stations studied here. The inclusion of gravitational settling is suggested to be required for a volcanic ash model.

# 1 Introduction

The European Monitoring and Evaluation Programme model developed at the Meteorological Synthesizing Centre - West (EMEP MSC-W) has been expanded to handle ash forecasting for the Norwegian Meteorological institute. Historically, the EMEP MSC-W Eulerian model has been used to deal with problems concerning acidifying substances deposition, and long-range transport of tropospheric ozone and particles (Simpson et al., 2012). The EMEP MSC-W model is already in use in a forecasting mode as one of the ensemble members of the MACC/CAMS daily ensemble production system for regional air quality forecasting (Marécal et al., 2015). This paper will present the developments of the EMEP MSC-W model that allow the model to describe transport of both gaseous and ash emissions from a volcanic eruption in both a forecast and hindcast setting; this version of the model is called the emergency EMEP (eEMEP) model.

The volcanic emission and transport of $SO_2$ can cause considerable air quality problems both close to a volcano and farther away. The preparation of the model for gaseous volcanic emissions is relatively simpler and we have documented the Holuhraun fissure eruption previously using the eEMEP model (Steensen et al., 2016). Volcanic eruptions that emit tephra into the atmosphere needs more consideration in the model compared to the standard setup of the model. Tephra are classified according to the particle diameter as blocks (< 64 mm) and lapilli (64 mm > d > 2 mm) that fall out quick and close to the volcano, the finer particles like coarse ash (2 mm > d > 64 μm) and especially fine ash (d < 64 μm) can stay in the atmosphere for days and be transported over large distances before settling to the ground. Exposed to fine ash, air traffic can experience both jet engine malfunction and damages to windshields (Casadevall, 1994). It is therefore of interest to study these smaller ash particles. The long-time closure of commercial air traffic during the 2010 Eyjafjallajökull volcanic eruption caused the European civil aviation authorities (CAAs) to step back from the previous zero-tolerance policy for air traffic in zones with observed or predicted ash and specific zones determined by ash concentrations with individual flight restrictions were introduced (UK Civil Aviation Authority, 2016). Currently there are three levels, low (< 2 mg m$^{-3}$) and medium (2 – 4 mg m$^{-3}$) ash concentration zones have lower restrictions and areas with high concentrations over 4 mg m$^{-3}$ are usually avoided. This change in policy requires higher accuracy in ash dispersion modelling, which is part of the motivation for the development of the eEMEP model.

There are different approaches for volcanic ash transport and dispersion models (VATDMs). Eulerian models such as the eEMEP model are computationally more demanding compared to Lagrangian models, which most Volcanic Ash Advisory Centres (VAAC) use, e.g. NAME (Jones et al., 2007) at the London VAAC, or HYSPLIT (Draxler and Hess, 1997) at the Washington and Anchorage VAAC. Other well known Lagrangian models used for ash dispersion are FLEXPART (Stohl et al., 2005) and PUFF (Searcy et al., 1998), the latter is also used as backup by the Washington and Anchorage VAAC. Some Eulerian models used for ash dispersion are MOCAGE (Josse et al., 2004) used at VAAC Toulouse, Fall3d (Folch et al.,



2009) and Ash3d (Schwaiger et al., 2012). The Eulerian models calculate the advection of ash at every grid point, and emissions are instantaneously mixed within the grid box. In particular peak concentrations are dependent on the grid resolution. Lagrangian models release tracers and calculate their trajectories, the mass loadings and concentrations are calculated from the number density of multiple releases of these tracers. This can lead to an uncertainty in regions with low particle concentrations, but the output resolution for Lagrangian models is independent of the resolution of the input data and can therefore be indefinitely high.

For all models, in addition to uncertainties caused by numerical diffusion and advection, uncertainties in the ash dispersion forecasting can also be due to imperfections of the meteorological driver. Initial conditions can only be set with a certain degree of accuracy when starting a numerical weather prediction model. The initial errors may amplify during the forecast and can result in forecast inaccuracies. In addition to these initial condition errors, there are uncertainties due to how the dynamics and physics are represented in the numerical weather prediction model (NWP). Ensemble forecasting was established in weather forecasting to estimate associated uncertainties by producing probability forecasts on the state of the atmosphere on the basis of multiple similar forecast runs with perturbed initial conditions or different model parameterizations (Palmer 2000, Iversen et al., 2011). Since 1992 ensemble forecast have been operational at both the National Meteorological Centre (NMC) (Toth and Kalany, 1993) and the European Centre for Medium-Range Weather Forecasts (ECMWF) (Palmer, 1993). Ensemble modelling has undergone large developments in recent years. In this study the eEMEP model will be run on state-of-the-art ensemble meteorology data on three different resolutions to see the different spread in dispersion.

The aim of this paper is to present the new developments and applications of the eEMEP model for describing the dispersion of volcanic emissions in the atmosphere. Both volcanic eruption examples with $SO_2$ emissions and ash are presented. At the start of an eruption $SO_2$ can act as proxy for ash (Thomas and Prata et al., 2011; Sears et al., 2013), and proven capability of modelling both ash and $SO_2$ can give increased confidence for dispersion of future eruptions. Section two describes the additions made to the model to improve the capability to simulate volcanic eruptions and bigger ash particles. There are several sources of uncertainties connected to the transport of volcanic emissions in a dispersion model. CPU efficiency of forecasts, uncertainties connected to the meteorological driver and numerical diffusion effects caused by changing horizontal resolution are studied by running the model with ensemble members for the first days of the Barðarbunga eruption in September 2014 in section 3. The practicality of ensemble forecasts is also evaluated. A short ash model validation is presented in section 4 with comparison to lidar observations. Uncertainties for the description of gravitational settling are studied by comparing two model simulations with and without this effect included. Summary and conclusions are given in section 5.



## 2 Model description

The standard EMEP MSC-W model is described in Simpson et al. (2012) and updates are in addition presented in the yearly EMEP reports as well as the updated model code (EMEP Status Report, 2016). The most important aspects of the standard model for volcanic emission dispersion is shortly described while new added components for the eEMEP are presented in
more detail, as well as how the model handles the source term and the operational setup.

### 2.1 Standard EMEP MSC-W model

Volcanic emissions are transported from the source by winds and lost due to several processes in the atmosphere. The advection scheme has a numerical solution based on the Bott's scheme (1989a,b), with the fourth order scheme in the horizontal directions and a second order version applied on the variable grid distances over the vertical resolutions. Time
steps used in the advection scheme are dependent on grid resolution. Winds and other meteorological parameters needed are given as input and the EMEP MSC-W model is adapted to run with output from several numerical weather prediction models. Horizontal resolution follows the meteorological driver, and model simulations with resolutions from very fine (few km) to low resolutions of 1x1 degree are possible. Temporal resolution of the meteorology input fields is typically 3-hourly. The model may calculate some field if they are not included in the data (e.g. 3-D precipitation or vertical velocity). The
chemical species, reactions as well as emissions included in the model have been developed over the history of the model, and the number of tracers is variable so the user can choose what to include. Deposition due to wet scavenging depends on precipitation fields given as input and specific removal efficiencies for the different gases and particle classes. Both in-cloud and sub-cloud removal are taken into account.

### 2.2 The eEMEP model

To improve the EMEP MSC-W model capabilities to model volcanic eruptions, the model was further developed in several components such that an efficient and flexible model framework was finally available for operations at the Norwegian Meteorological Institute. This emergency model is simplified in parts with respect to the original EMEP MSC-W model to be computational more efficient.

**Meteorological driver**

On a day to day basis the eEMEP model use ECMWF forecast meteorology, pre-processed for the CAMS 50 chemical weather forecasting at a resolution of 0.25 x 0.125 degrees latitude longitude. More details have already been provided in Steensen et al. (2016). For this study the eEMEP model has also been setup to run on ensemble weather forecast data as demonstrated in section 3. Ensemble forecast require considerable computational time. Explosive volcanic eruptions may in some cases inject ash and gases at heights well above the tropopause. This required that the eEMEP model, depending on
actual eruption conditions, provides the possibility of introducing additional vertical levels to achieve increased vertical resolution as well as a higher model top. The standard EMEP MSC-W model has 20 vertical levels reaching up to 100 hPa,



at around 16 km. For specific volcanic eruptions, where the ejection force places the emissions higher in the atmosphere, a more flexible version was developed with hybrid eta levels, where meteorological data from additional vertical levels, that are available in the eg. ECMWF driver model, can be used. Model simulations presented in this paper are done with 40 and 42 vertical levels. Vertical levels close to the surface where not altered because this would have changed the well

characterized surface exchange processes in the EMEP model.

**Volcanic source**

A specific volcano source module reads in volcanic emission parameters from a file containing ash flux (kg s$^{-1}$), height interval in which emissions are injected along with a time line of release intensity. When only an emission top height is given the emissions are distributed uniformly down to the height of the top of the volcano. If more detailed information is

provided the emissions are spread uniformly over the model vertical levels that are closest and within the height interval given in the input file. More sophisticated plume descriptions such as output from plume models that use atmospheric conditions like PLUMEIRA (Mastin, 2007) or emission profiles calculated through inversion techniques (Stohl et al., 2011) can therefore be used as input to the volcano source module. Fine-ash particle sizes and the distribution over the size bins can also be changed to what is provided by the source term. If the source term denotes all the tephra released from the

volcanic eruptions, the largest sizes of tephra that quickly settles to the ground are excluded by using a fine-ash fraction either as given in Mastin et al. (2009) for the specific volcano or as provided based on more up-to-date case specific information.

**Gas chemistry**

To perform quick simulations of the spread of volcanic emissions, sophisticated chemistry and trace species emissions are

computationally too demanding and the eEMEP model has been configured such that they are excluded. For volcanic eruptions with SO$_2$ emission the sulphate production is added. More detail can be found in Steensen et al. (2016).

**Ash properties and removal processes**

Apart form the wind advection of volcanic ash and wet scavenging as described in the standard EMEP MSC-W, an important process for the simulation of volcanic ash is the gravitational settling. In the standard EMEP MSC-W,

sedimentation and dry depositions of the different pollutants are only calculated in the lowest model layer. Fine ash is large enough to have an effect from gravitational sedimentation and is emitted higher in the atmosphere compared to other coarse aerosol like sea salt and desert dust. A module that calculates gravitational settling in all vertical levels for ash particles is implemented. The assumed terminal fall speed $v_s$ for the ash particles are set as

$$v_s = \sqrt{\frac{4g(\rho_p - \rho_a)d}{3C_d\rho_a}}, \tag{1}$$

where $g$ is the gravitational constant, $\rho_a$ and $\rho_p$ are the densities for air and ash particle respectively. Default density for ash are assumed 2500 kg m$^{-3}$, however the density can range from 700 kg m$^{-3}$ for the most porous part of tephra to 3300 kg m$^{-3}$





for crystals (Wilson et al. 2011). $d$ is the particle diameter and $C_d$ is the drag coefficient. Wilson and Huang (1979) present a drag coefficient as a function of the particle shape found from fall velocity measurements of ash particles.

$$C_d = \frac{24}{R_e} F^{-0.828} + 2\sqrt{1.07 - F} \text{ , where } R_e = \frac{v_s \rho_a d}{\eta_a} \text{ ,} \qquad (2)$$

$R_e$ is the Reynolds number and $\eta_a$ is the dynamic viscosity of air. $F = (a + b)/2c$ is a shape factor and $a < b < c$ are the
three principal diameters of the particle. Although ash particles can vary in shapes the default value of F assumes that the ash particle is close to spherical at 0.8. The smallest fine ash particles can in some circumstances be of similar size as the mean free path length of an air molecule ($\lambda_a$), if this occurs the particle are in a slip-flow regime and non-continuum effects has to be taken into account. The vertical fall velocity is therefore modified with the Cunningham slip-factor and the Knutsen-Weber term (Jacobson, 1999, eq. 16.25). Fine ash is shown to fall faster than the Stokes Law calculates (Rose and Durant,
2009), therefore the more spherical shape (F=0.8) is set as default since slip flow corrections increase the fall speed for fine ash for more spherical fine ash particles (Schwaiger et al., 2012).

**Operational set-up**

The eEMEP model runs operationally every day at the Norwegian Meteorological Institute, for dispersion scenarios of volcanic emissions as defined in Mastin et al. (2009) for four selected volcanoes in the region of interest. If an increased risk
for an eruption is given for any volcano, one or several of the default volcanoes are replaced with the volcano at risk of eruption. Meteorological input data are available every day before 08 30 UTC and 20 30 UTC and forecasts starting from 00 UTC and 12 UTC are run from these respectively. A standard eEMEP model simulation takes less than half an hour making forecasts from 00 UTC and 12 UTC available before 9 UTC and 21 UTC.

In case of a real volcanic eruption, several simulations are used and started as shown in Figure 1. The purpose of the every-
day initial forecast with a default volcanic source is to provide a conservative first estimate of the dispersion. However, because of the high uncertainty in source intensity and vertical profile as well as ash size distribution the resulting concentrations are very uncertain. Thus, as soon as possible, source receptor model simulations, with a unit emission (1 kg s$^{-1}$) released every third hour over multiple emission heights, are started that are used as input data for a source inversion calculation (Stohl et al. 2011), shown in Figure 1 as thick arrows. The goal of applying the inversion algorithm is to create a
source term that causes the model simulation to be more similar to observations. A very early timing of these model simulations is not as crucial since the inversion algorithm is dependent on good satellite observations to constrain the solution, meaning that enough satellite observations at some distance from the volcano are required. The source receptor simulations run for a 24 hour forecast to not include further uncertainties longer into the forecasts, and are restarted the next day for another 24 hour cycle. Model simulation with the inversion-derived emission estimate (dashed lines) are expected to
be ready before a new forecast meteorology input dataset is available, since the inversion calculation is not computational demanding. More details on using the inversion method in a forecast setting are given in Steensen et al. (submitted to ACP, 2016). The length of the source receptor model simulations are set here as an example of three days as the inversion study





found that additional satellite observations are seen to have little effect on the emission term after 48 hours. The ash may however stay in the domain for a longer time so the model simulations with the optimized emission term have to be started from an earlier time.

## 3 Meteorological predictability of volcanic plumes, example SO$_2$ from Barðarbunga

The EMEP MSC-W model results have been compared to model results from other dispersion models and observations in several studies. In particular Steensen et al. (2016) compares model simulations for the Barðarbunga eruption to satellite and ground observations of SO$_2$ and SO$_X$ wet deposition. A simplified emissions term is used where SO$_2$ is released with a constant rate, uniformly distributed over three emissions heights, to see which height produces a simulation that matches better with observations. With some discrepancies caused by the description of the planetary boundary layer also found in

Schmidt et al. (2015) with the Lagrangian NAME model and in Boichu et al. (2016) with the Eulerian CHIMERE model, the EMEP MSC-W model matches well the observed surface SO$_2$ timing and concentration levels. Compared to SO$_2$ column satellite observations from the Ozone Monitoring Instrument (OMI) similar mass loadings and dispersions patterns are found. However, it remains a question how much the uncertainty in the meteorological fields determines the quality of the volcanic plume predictability.

Ensemble forecasts consist of several almost identical simulations to quantify the uncertainty of a forecast. Large spread between the ensemble members caused by a large difference between possible future scenarios indicates a high uncertainty in the forecast. Combining the eEMEP model with ensemble forecasts would create an opportunity for quantifying the uncertainty in the eEMEP ash/SO$_2$ forecasting related to the uncertainty in the meteorology. However, running the eEMEP model on several (tens of) ensemble forecast members (on a relatively high resolution) is computationally a very expensive

task. At the same time, the volcanic emission forecasting system is required to deliver results very fast and several times a day. One might think at first, that this leaves us with the choice between a low resolution ensemble forecast and a high resolution deterministic forecast for the operational volcanic emission forecasting system.

Here we investigate how the different resolutions of the ensemble forecasts affect the spread of the volcanic plume for SO$_2$ for three days in the beginning of the Barðarbunga eruption, including the probability of the resulting SO$_2$ concentrations in

the different members to be over different thresholds. Although this part focuses on volcanic emissions of SO$_2$, similar results may be expected for spread of the fine-grained long-range transported volcanic ash.

### 3.1 Model setup

The eEMEP model is run on meteorological ensemble forecast data from the Grand Limited Area Ensemble Prediction Systems (GLAMEPS) for the Barðarbunga eruption case. The starting dates, 3 to 5 September 2014 - from which respective

48 hour forecast are launched - , correspond to the first phase of the Barðarbunga volcanic eruption.



Significant amounts of $SO_2$ were ejected into the atmosphere during the Barðarbunga eruption, but little ash. Schmidt et al. (2015) studied the emission term during September by comparing model simulations to satellite data, and found that a $SO_2$ emission estimate with a 120 kt d$^{-1}$ flux over an eruption column between 1500 m to 3000 m matched best for the first days of September. This emission term is also supported by Thordarson and Hartley (2015) and used here.

**GLAMEPS meteorological data**

GLAMEPS aims to account for all the major sources of weather forecast inaccuracy by looking at both the differences due to model parameter uncertainty and initial state perturbations (Iversen et al., 2011). GLAMEPS ensemble forecast is produced at ECMWF, and in 2014 the ensemble consisted of 50 members from both the HIRLAM (High Resolution Limited Area Model) and ALADIN (Aire Limitée Adaptation Dynamique Développement International) model. To include the uncertainty

in the forecast, members are perturbed both in the initial field and on the model domain border. The perturbations are from the EuroTEPS (European Targeted Ensemble Prediction system), a version of the global ECMWF EPS, with higher resolution on a smaller European domain (Frogner and Iversen, 2010).

This study will only use the 24 HIRLAM (High Resolution Limited Area Model) perturbation members of the ensemble, (not the control member). The 24 HIRLAM members are split between two different cloud physics parameterisations.

HirEPS_S members use the STRACO scheme (Sass et al., 1999; Undén et al., 2002) for stratiform, convective cloud, and precipitation, HirEPS_K members uses the Kain-Fritsch schemes for deep cumulus (Kain and Frisch, 1990; Kain, 2004; Calvo, 2007) and Rasch and Kristjansson (1998) for stratiform clouds and precipitation (Ivarsson, 2007). To also include the uncertainty in the forecast caused by the start time of the forecast, members are divided in two groups with two different forecast start times. Six members of the HirEPS_S and six of the HirEPS_K start the forecast at 00 UTC and 12 UTC, and

the remaining 12 of the ensemble members start the forecast at 06 UTC and 18 UTC. Each member has a forecast time of 72 hours. The original resolution is 10x10 km. Such a fine resolution is computationally very demanding.

The GLAMEPS data have been downloaded from ECMWF for the period from 3 to 5 September 2014, corresponding to the first phase of the Barðarbunga volcanic eruption. Each member is used as input data for the eEMEP model to run 48 hour forecasts starting from 00 UTC from each of the three days by using the 18 UTC and 00 UTC meteorological forecasts. That

means that for half of the members, the forecast is six hours old (the forecasted started 18 UTC). The relatively short forecast of 48 hours is chosen due to the large uncertainties related to the emission term when running a forecast of volcanic emission (VAAC London only issues a maximum 24 hour forecast). Furthermore, running the full 72 hour forecast is not feasible due to the different start times of the forecasts (18 UTC and 00 UTC). In contrast to what is possibly done for a real case, all the forecasts are started from an $SO_2$ free state and not restarted from the previous forecast simulation. This is done to

investigate only the difference in spread due to the different weather situation over the three days studied, and not take into account possible differences in the initial $SO_2$ field.

Three different horizontal resolutions are generated as input, the original high resolution of 10x10 km, a medium resolution of 20x20 km and a low resolution of 40x40 km, referenced hereafter as high_res, mid_res and low_res. High resolution NWP is desired as more processes in the atmosphere can be resolved. A simple reduction in resolution of the meteorological





input data is obtained here by letting every other and every fourth original grid value represent a twice and four times bigger grid point respectively. Alternatively, one could have aggregated 10x10 km grid cells into 20x20 km and 40xkm grid cells. This would lead to a smoother field, but the largest values/variations would have been lost. The GLAMEPS domain covers an area from 16 to 81 degrees north and 67 degrees west to 83 degrees east which means that Europe and the North Atlantic are well covered including some parts of Sahara and Newfoundland. There are 40 vertical levels for all the three horizontal resolutions. Table 1 lists all simulations in this paper.

## 3.2 Results

The spread in the ensemble forecast of $SO_2$ is presented here by calculating the possibility of an ensemble member to be over certain threshold values. Figure 2 and 3 show the frequency of ensemble members over a low 5 DU (Dobson Unit) threshold and a high 50 DU threshold after 48 hours forecast, respectively. A high and low threshold is chosen as increased dilution will lead to a larger area with lower concentrations and a smaller area with higher concentrations. For the first forecast starting 3 September 00 UTC, the $SO_2$ emitted from the volcano is caught in a low pressure system and transported over Northern Scandinavia and Russia. In the forecast simulations starting 24 and 48 hours later, the transport of volcanic emissions is southward over Great Britain by a high pressure system placed more southerly compared to the first system. Compared to the two higher resolution forecasts, the low_res forecasts have a large area where 20 or more members agree and have column loads over the 5 DU threshold after 48 hours of dispersion. The mid_res and high_res simulations show however a larger spread between the members with a bigger area with only one or a few members above the low threshold. This larger spread among the higher resolution forecast is also shown in the Figure 3. The high_res forecast have members that exceed the 50 DU threshold far away from the source, as seen over the coast of Northern Russia in the first forecast (0309 00 UTC + 48 hours) and also further down in the high pressure system appearing during the two later forecasts. Due to increased numerical diffusion in the low_res forecasts, the area where the members are over the higher threshold is much smaller and mostly confined to an area close to the source.

The difference in the spread is also seen to be weather dependent especially when using a low threshold. For the first forecast the volcanic $SO_2$ is transported over longer distances and the small low pressure system is positioned differently in the ensemble members. The two forecasts started later experience weaker northerly winds and smaller plume position differences appear in between the members.

To further investigate the differences in the three resolutions, Table 2 shows the area summed up at time step +48 hours for all the three forecasts where simulated $SO_2$ column loads exceed the given threshold values. There is a 34% larger area with column burdens above 0.25 DU in the low resolution forecasts compared to the high resolution forecasts, due to the increased diffusion. At 10 DU, the difference in area between the three resolutions is minimal, while for the thresholds above the 10 DU the high resolution forecast exhibits a larger area.

Figure 4 shows the frequency above 10 DU of low and high resolution averaged over the hours from 8 to 16 UTC on 5 September, for the forecast starting 4 September 00 UTC. The model results are compared to OMI (Ozone Monitoring



Instrument) satellite observations from overpasses during the same time. Retrievals are described in Theys et al. (2015) and have an assumed plume height of 7 km, which is higher than the actual plume height and as a consequence the retrievals have too low values. Even though the column burdens from OMI and the model results are not easily comparable (see discussion in Steensen et al., 2016), the patterns should be similar. The satellite has high concentrations going south from

Iceland in a thin filament. Even though the total amount of area, where ensemble members show $SO_2$ column loads above the 10 DU threshold, is found equal for the three resolutions, the high_res ensemble forecasts have members exhibiting further south loads over the threshold. An area in the southwest is not captured by either of the forecasts. Since forecast starts with no $SO_2$ emitted before 00 UTC 4 September, we believe this area is affected by older emissions and thus not apparent in our model calculations.

The higher resolution ensemble members show higher concentrations further away from the volcanic eruption site in more narrow plumes that resemble the observed plumes more. However, the location of the plumes varies more between the high resolution ensemble members, e.g. a larger spread between the ensemble members indicates a higher uncertainty. Although a lower uncertainty in a forecast is appreciated as it indicates that the weather situation is stable, less spread in low resolution also indicates that information in the meteorology is lost when reducing the resolution. This suggests that running ensemble

forecasts with low resolution for transport of volcanic emissions is less meaningful as it would not show the actual uncertainty in the forecast.

Even though a high resolution is desirable, the computational efficiency is important in an emergency forecast environment. For this study, the highest resolution runs use over 13 times more computational time than the lowest resolution runs, while the mid_res simulations use only five times more.  To run a total ensemble forecast with high resolution for volcanic

eruptions may therefore not be feasible. From a pragmatic point of view, ensemble forecasts for volcanic emissions are most valuable in situations where the weather forecast is uncertain. Thus an alternative would be to launch ensemble forecasts only in unstable weather situations (as predicted by the ensemble weather prediction models). This study indicates that less information is lost between the high_res and mid_res than going from the mid_res to the low_res resolution, suggesting that resolutions around 20x20 km could be a reasonable choice for ensemble (and deterministic) forecasts when needed.

**4 Ash model testing and validation**

**4.1 Model setup**

The eEMEP model with improved ash modelling capabilities as described above is tested here for the Eyjafjallajökull eruption in 2010. For this purpose the model is run with the emission term from Stohl et al. (2011), an emission term constrained by satellite observations through an inversion routine. The ash is distributed over nine size bins from 4 µm to 25

µm and the density is set by default to 2500 kg m$^{-3}$. Our eEMEP model uses here meteorological data from ECMWF, namely the IFS (Integrated Forecasting System) model with a 0.25 x 0.25 degree latitude longitude resolution, selecting 42 of the



lower levels from the original IFS 60 vertical levels, setting the eEMEP model top to around 30 km. The simulations are performed for the main volcanic ash eruption episode from 13 April.-25 May 2010.

## 4.2 Comparison with other ash models

Part of the validation has been done in the scope of the Norwegian ash project and shall not be repeated here in all detail. We compared initially ash dispersion from this eruption calculated with eEMEP and FLEXPART model results as well as the Norwegian Meteorological Institute version of the NAME model SNAP (Saltbones et al., 1994), and found very similar ash plumes in all three models (Norwegian ash project, 2014). Figure 5 shows results where all three model results are compared to satellite ash retrievals from SEVIRI (Spinning Enhanced Visible and Infrared Imager) and IASI (Infrared Atmospheric Sounding Interferometer) available from vast.nilu.no. The FLEXPART model is used in several studies and validated towards observations for several volcanic eruptions with ash emission (Stohl et al. 2011; Kristiansen et al. 2012; Moxnes et al 2014). All models use the same wind fields from ECMWF, and the conclusion for this eruption case was that neither the Eulerian nor the Lagrangian models showed particular better performance. The structure and intensity of the plumes was rather similar, reproducing the observations fairly well when it comes to the ash fields.

## 4.3. Validation with lidar observations

Apart from the horizontal dispersion, the vertical placement of the transported ash may have important consequences for impact assessments, both for air quality and air traffic perturbations. Meteorological processes such as subsidence and frontal lifting may alter the initial vertical distribution of ash. In addition ash removal and settling may alter the vertical distribution. Although several observational sets are available for the Eyjafjallajökull eruption, to test here the treatment of gravitational settling for ash particles in the eEMEP model, model results with and without gravitational settling of ash  included are compared to lidar observations of the ash layer.

Lidar observations provide a vertical location of aerosol. The European Aerosol Research Lidar Network (EARLINET) consisted at the time of the Eyjafjallajökull eruption of 27 aerosol stations over Europe. On 15 April, an alert was given to start continuous measurements providing, if weather conditions permitted, an hourly vertical coverage of the ash cloud over Europe (Pappalardo et al. 2013), documented as a consolidated dataset which we use here. Ash is detected as significant aerosol backscatter signal, linked to the Iceland eruption through backward trajectory analysis. Only backscatter profiles with a relative statistical error from signal detection less than 50 % are used to retain a reliable aerosol mask. The vertical resolution in the dataset ranges between 60 and 180 m for the different stations. The dataset includes the identified top and bottom of the ash layer, as well as the centre of mass, the altitude where most of the aerosol load is located. Identified ash layers where other aerosol sources are also found from e.g. continental aerosol are classified as mixed layers.  These mixed layers are also given with the maximum and minimum observed height and centre of mass. Observed planetary boundary layer (PBL) height is also included in the database. The six lidar stations used here are situated in Central Europe (see Figure 6), covering coastal stations, inland and mountain regions. Weather conditions at the lidar stations, and sometimes technical





issues, made it difficult to continuously produce observations. For example, frequent low clouds over Cabauw prevented most lidar retrievals there. Observations at Neuchatel are also limited to the first episode in April. Altogether, the ash layer was observed over a long period over central Europe during the Eyjfjallajökull eruption and as the ash has been transported over a long distance the effect of gravitational settling may be visible, making this dataset the best available at the time for
our purpose.

Figure 7 shows the model concentrations for the simulation with gravitational settling over the entire Eyjafjallajökull period along with observed height of the ash layer and height of the mixed aerosol layer at the EARLINET stations. Although the mixed layers may be weighted with the other aerosol they are plotted here also. Figure 8 concentrates on the centre of mass comparison (without the mixed layers).

Ash was first detected at the Hamburg station during the morning of 16 April, 48 hours after the start of the eruption. Ash was also observed early at the other stations, and while the timing of the observations match well at Hamburg and Leipzig, at Neuchatel ash is observed before the model has transported ash to this station. At Cabauw, the first part of the ash plume is not detected by the lidar, while the second part shows similar simulated and observed level of maximum concentrations. A descent in the ash plume is observed at several of the stations, at Hamburg between 05 UTC to 17 UTC 16 April and at
Palaiseau, Munich and Neuchatel stations from 16 UTC 16 April to 00 UTC 17 April (Pappalardo et al. 2013). This decent is due to the high pressure system that transported the ash over Central Europe. The model shows this shift in model height from the higher first part of the plume to the lower second part of the plume for all the stations. At Leipzig, Hamburg, Palaiseau and Munich much of the simulated ash is even below the observed PBL in the night of 17/18 April. As the ash layer in EARLINET is calculated above the observed PBL, observed centre of mass is at higher altitudes than the model ash.

The ash cloud persisted over Central Europe until 26 April, model and observed ash layer are mostly at similar heights for this later April period. There are some discrepancies: for Cabauw the observations vary in height over short time periods indicating uncertainties for these measurements, and in Munich, the maximum heights of the observed ash layer are at higher altitudes than the model maximum ash height, while the observed centres of mass are close to the observed lowest layer of ash, indicating that most of the ash mass indeed is lower in the atmosphere and more comparable to model heights. In
Leipzig a few observations of centre of mass on 16 and 18 April are much higher than the model centre of mass heights and the corresponding heights at the other stations at this time, signifying that there may be uncertainty in these observations.

From May 2 the model results show small ash concentrations at the lidar stations, due to small ash emissions after 29 April. On May 5-6 ash is observed lower down in the atmosphere compared to simulated ash at Hamburg, however a layer where ash is mixed with other aerosol is detected at higher altitudes more similar to where the model has ash. More ash was then
emitted on 5 May (Stohl et al 2011), but southerly winds transported the ash over Spain and the Atlantic Ocean. Not until the night 16/17 May are weather conditions favourable again for transport of ash to central Europe. No measurements are available for this time at Neuchatel. The other stations have observations of the ash layer at similar altitudes as the model.

To show more broadly the impact of the gravitational settling processes on the vertical profile of ash, Figure 8 shows all calculated centres of mass for ash in the model simulation with and without gravitational settling. The rather small



displacement between the two model simulations implies that not gravitation but rather weather and emission height are the main driver for the ash layer height. This is especially visible in the simultaneous rapid decrease in centre of mass height for the first plume (17-18 April) in both simulations. On some occasions there are larger differences between the two model simulations, specifically in the beginning of May during a period with smaller concentrations. Unstable north-westerly winds at this time can cause the small differences in height distribution of ash to grow over time due to different wind directions in the column.

In order to compare to the observed values more properly, a centre of mass above the observed PBL is calculated for the two model simulations (only for the cases when an observed height is available), see Figure 8. Model centre of mass are generally lower than observed altitudes for both model simulations, indicating that the model simulations have too much descent of the ash layer eg. around 18 April, independent of inclusion or not of gravitational settling. Figure 9 show scatterplots where the observed ash centre of mass height (not including the mixed layers) is plotted against model with and without gravitational settling at the stations. As discussed above some measured and modelled values are unrealistic high, therefore only values below 8 km are taken into account for correlation calculations. The scatterplot confirms that observed heights are generally higher than model calculations. At Palaiseau and Hamburg, model height descends faster than observed on 20 April causing the low correlation at these stations. Neuchatel generally exhibits higher observed centre of mass, explaining possibly a slightly higher correlation for the model simulation with no gravitational settling. Except for Neuchatel and Hamburg however, the model simulation with gravitational settling exhibits a slightly higher correlation to lidar retrieved height data compared to the model simulation without gravitation.

## 5 Summary and conclusions

A new model version of the standard EMEP MSC-W model has been developed, aimed at modelling dispersion of volcanic emission, called the eEMEP model. Changes with respect to the standard model are: a simplified gas chemistry; a modification of the aerosol part to handle ash particles in different size classes; the description of gravitational settling of ash particles; a volcanic source module which has a default source term and can be altered to include improved source estimates; an increase in vertical levels to increase the model top and vertical resolution; the possibility to run as an ensemble model based on ensemble meteorological forecasts; a formal procedure for an operational use of the model in an emergency case and an inversion algorithm coupled to the model, using satellite data to retrieve an improved source estimate (see Steensen et al. 2016, submitted to ACPD). With this model version we document here selected important aspects of the volcanic gas and ash dispersion simulation.

We have first studied the impact of ensemble meteorological input fields of different resolution on the dispersion of volcanic emissions from Iceland. Compared to Lagrangian VATDMs, Eulerian models such as the eEMEP model have inherent numerical diffusion dependent on the grid size. eEMEP model simulations thus have to have a sufficiently high resolution, especially when peak concentrations shall be predicted, for example for the purpose of establishing flight restriction zones.





High resolution simulations are however computationally demanding while obtaining results quickly is critical in situations with volcanic eruptions. How to best use CPU resources for transport of volcanic emission is studied here by looking at the change in spread between ensemble model simulations on three different resolutions. The eEMEP model is run for a 48 hour forecast from three start dates for the Barðarbunga eruption period with meteorological fields from 24 HIRLAM ensemble

members originally produced for the GLAMEPS forecast. The original 10x10 km resolution is degraded to lower resolutions of 20x20 km and 40x40 km.

The increased numerical diffusion causes a larger area (+34%) to be covered by the volcanic tracer in the low resolution simulations than in the high resolution ones. For the higher resolution forecast simulations, the members show more spread between them and there are members with higher concentrations further away from the source. Therefore there is also a

greater possibility for a member to exceed a high threshold concentration. For all three simulations, the spread between members is seen to be weather dependent and a measure for how uncertain the forecast is. The increased agreement between low resolution ensembles due to increased numerical diffusion limits the importance of the ensemble forecast. Ensemble forecasts have to be done with sufficient high resolution to show the real uncertainty of the weather situation. Compared to satellite observations, the high resolution model simulations also match better the transport of a narrow volcanic $SO_2$ cloud.

High resolution ensemble forecast may nevertheless not be possible due to the computational cost. The study shows that there is a bigger change in transport when going from 40x40 km to 20x20 km resolution compared to 20x20km to 10x10km, indicating that the increased cost to run a high resolution simulation at 10x10km may not give the same increase in quality of the result. The study also shows that low resolution simulations are sufficient for quick results to predict the most likely transport of volcanic emission, while high resolution model simulations are needed to estimate possible occurrence of high

peak concentrations.

The vertical dispersion of ash transport was studied. Gravitational settling for ash tracers is added in the model over the entire vertical column. This addition is evaluated by comparing a model simulation with and without gravitational settling to observations during the 2010 Eyjafjallajökull eruption. EARLINET ground stations measured the vertical location of the volcanic ash layer over the eruption providing hourly observations of the height and centre of mass for the ash layer when

the weather allowed it. Centre of mass calculated for the two model simulations show that gravitational settling displaces the centre of mass closer to the ground by up to 1km. Besides emission height the weather situation is found to be a more important factor than gravitation for the height of the ash layer as most of the vertical displacement is caused by subsidence in high pressure systems and is similar in both model simulations. An example is a rapid descent in ash plume height on 16 April caused by an anti-cyclone seen in both observation and model. However the descent in the model is quicker and puts

the ash closer to ground compared to observations, especially at Hamburg and Leipzig lidar stations. A second descent in the ash layer at the stations is seen 20 April, and this subsidence occurs later in the observational data at Hamburg and Palaiseau compared to the model data. The model has a centre of ash mass height on average below the observed one, independent on gravitational settling. Calculated correlation between observed centres of mass height and corresponding model heights are higher in the model simulation with gravitational settling for four of the six stations studied here, suggesting improved



quality of model when including the gravitation process. The addition of gravitational settling is found to have a relatively small influence on the vertical placement of the ash layer and thus is responsible only for a small improvement in model results.

Even with the included gravitational settling in the EMEP model, the assumed density, shape and size distribution of the ash
particles bring along large uncertainties during a forecast situation. Ash properties show large differences in between volcanic eruptions (Vogel et al. submitted to JGR 2016). The Eyjafjallajökull model results presented here are initiated with a time and height resolved emission estimate calculated by inversion with FLEXPART model results, constrained with satellite observations (Stohl et al. 2011), to be used with the eEMEP model for a new volcanic eruption in an operational setup. Uncertainties in satellite retrievals due to meteorological clouds that obscure ash clouds and a 0.2 g m$^{-2}$ ash cloud
detection limit (Prata and Prata, 2012) need to be explored further. Low concentrations in satellite and model data are in particular uncertain, as shown during the intermediate period (end of April and beginning of May) where only small emissions were released and then observed at the lidar stations as almost insignificant ash clouds. Another factor of uncertainty is the variability in ash particle sizes, which changed between the first eruption period in April and the second in May (Dellino et al. 2012). Both model and satellite retrievals used as input to the inversion, operate today with assumed
equal ash characteristics during the whole eruption period. Other processes, such as fine ash aggregation that increases the gravitational settling speed and reduces the atmospheric residence time (Brown et al. 2011), were also observed during the Eyjafjallajökull eruption (Taddeucci et al., 2011), but are not included in the model at this time.

Although a correct model description of bulk volcanic emissions is useful, other factors such as model resolution, details of the source term and the model set-up are seen as important for safety assessments. The developed model is capable to guide
near real time emergency assessments of the spread of high volcanic gas and aerosol concentrations.

## 6 Code availability

The model code to the standard EMEP MSC-W model is available on gitbub: https://github.com/metno/emep-ctm. The eEMEP model with reduced chemistry and gravitational settling for ash is available from here: https://github.com/metno/emep-ctm/releases/tag/rv4_10_eEMEP_ASH

**Acknowledgement**

The authors would like to thank Fred Prata and Lieven Clarisse for SEVIRI and IASI satellite data for the model comparison and Nina I. Kristainsen for numerous inspiring discussions and technical help. We also thank Inger-Lise Frogner for the valuable help with obtaining GLAMEPS data. We thankfully acknowledged the EARLINET lidar data providers for assembling a very useful lidar dataset. This work has also received funding under the ACTRIS project from the European
Union's Horizon 2020 research and innovation programme under grant agreement No 654109. The work done for this paper



is funded by the Norwegian ash project financed by the Norwegian Ministry of Transport and Communications and AVINOR. Model and support is also appreciated through the Cooperative Programme for Monitoring and Evaluation of the Long-range Transmission of Air Pollutants in Europe (No: ECE/ENV/2001/003). This work has also received support from the Research Council of Norway (Programme for Supercomputing) through CPU time granted at the super computers at
NTNU in Trondheim.

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

Table 1: List and names of model simulations used in this paper.

| Name | Period | Meteorology | Resolution | Emission |
|------|--------|-------------|------------|----------|
| Low | 03.08.2014 00UTC + 48 UTC<br>04.08.2014 00UTC + 48 UTC<br>05.08.2014 00UTC + 48 UTC | GLAMEPS 24 perturbed ensemble members | 40 km x 40 km | Schmidt et al. (2015) |
| Mid | 03.08.2014 00UTC + 48 UTC<br>04.08.2014 00UTC + 48 UTC<br>05.08.2014 00UTC + 48 UTC | GLAMEPS 24 perturbed ensemble members | 20 km x 20 km | Schmidt et al. (2015) |
| High | 03.08.2014 00UTC + 48 UTC<br>04.08.2014 00UTC + 48 UTC<br>05.08.2014 00UTC + 48 UTC | GLAMEPS 24 perturbed ensemble members | 10 km x 10 km | Schmidt et al. (2015) |
| Eyja_grav | 13.04.2010 – 25.05.2010 | ECMWF IFS | 0.25 x 0.25 degree | Stohl et al. (2011) |
| Eyja_no_grav | 13.04.2010 – 25.05.2010 | ECMWF IFS | 0.25 x 0.25 degree | Stohl et al. (2011) |

Table 2: Total area A where $SO_2$ loads are above a given threshold, found on average in members of the low, mid and high resolution ensembles (see table 1). Areas are accumulated at the end of a 48 hour forecast, corresponding to Fig. 1 and 2.

| $SO_2$ load threshold [DU] | $A_{Low}$ [$1e^6$ km$^2$] | $A_{Mid}$ [$1e^6$ km$^2$] | $A_{High}$ [$1e^6$ km$^2$] |
|---|---|---|---|
| 0.25 | 69.7 | 58.3 | 52.1 |
| 5 | 25.0 | 23.1 | 21.3 |
| 10 | 14.9 | 14.8 | 14.3 |
| 20 | 7.4 | 8.0 | 8.1 |
| 50 | 1.9 | 2.4 | 2.2 |





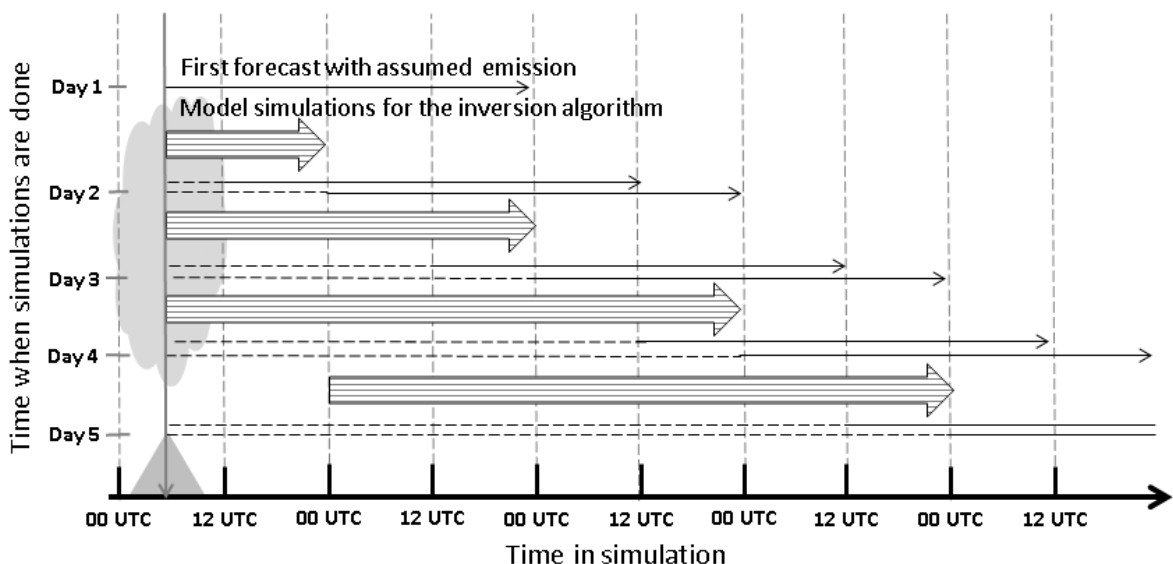

**Figure 1: Sequence of model simulations started at the Norwegian Meteorological Institute in the case of a volcanic eruption. The single black arrows indicate a 48 hour forecast simulations. The thick striped arrows represent the multiple model simulations started for the inversion algorithm to retrieve an improved emission estimate using satellite data. Dashed lines represent spin-up model simulations with emission estimate found by the inversion algorithm. These model simulations are continued as forecasts (single black arrows). The chronological order of simulations starts from the top, so new forecast results are available every 12 hour. The inversion simulations are restarted every 24 hour.**





**Figure 2: Map of number of ensemble members that is locally exceeding a 5 DU SO$_2$ limit. Counted after 48 hours in the low, mid and high resolution ensembles, in the left, middle and right column respectively, for start time 00 UTC 3. September (panels a,b,c), 00 UTC 4. September (panels d,e,f) and 00 UTC 5 September (panels g,h,i).**





**Figure 3; The same as Figure 2, counting ensemble members locally exceeding a 50 DU limit.**



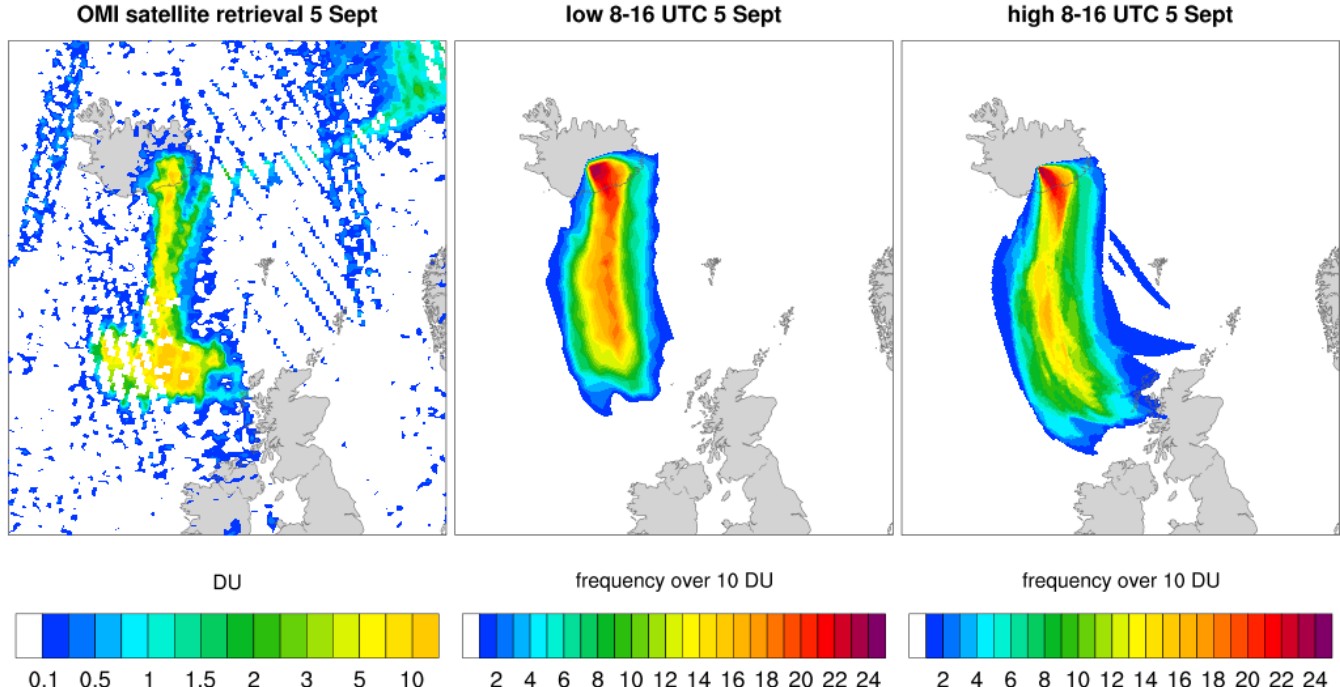

**Figure 4; OMI retrieval of SO₂ (left) for the satellite overpasses between 8 UTC and 16 UTC 5 Sept. Frequency of ensemble members over 10 DU for the low (middle) and high (right) resolution runs. Frequencies are computed every hour and averaged over the same time period, using the forecast runs started 4 September 00 UTC.**





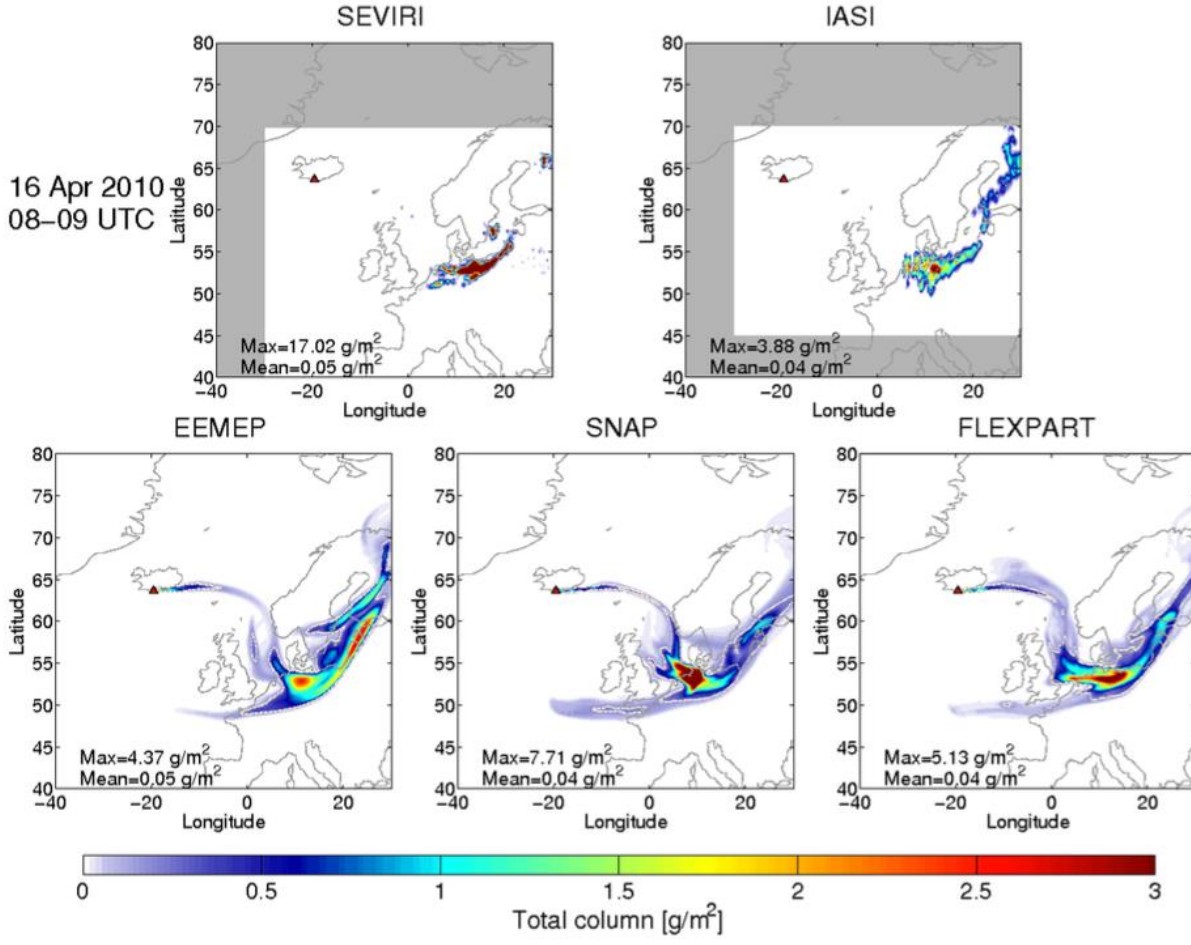

**Figure 5: Mean ash column burdens from 8 to 9 UTC 16 April for SEVIRI and IASI satellite ash retrievals, and eEMEP, SNAP and FLEXPART model simulations.**



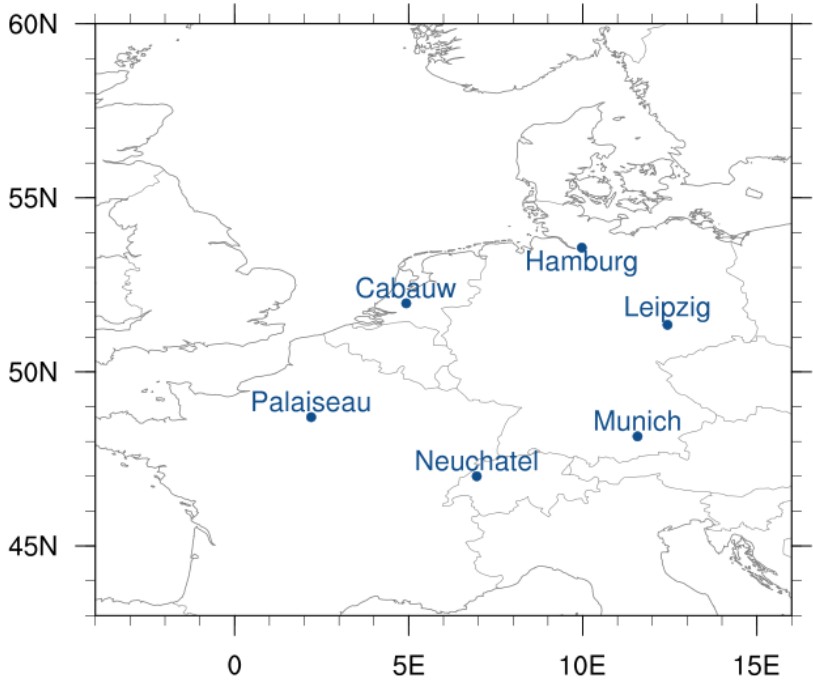

**Figure 6: Map of EARLINET lidar measurement sites used in the study.**





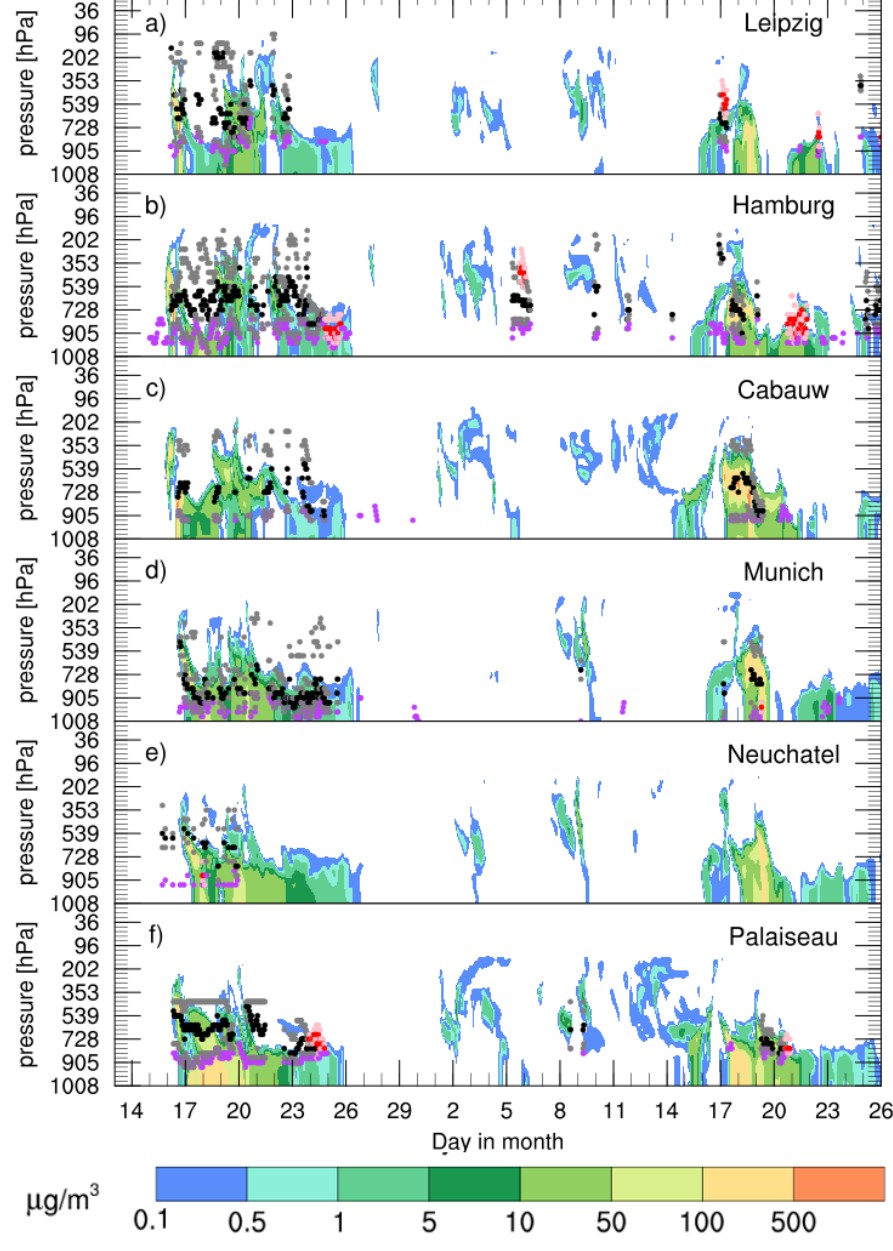

**Figure 7; Height-time profiles of ash concentrations from eEMEP model, including gravitational settling, at the six EARLINET lidar stations (see figure 6) in April-May 2010 episode (contour graph in background). Lidar-detected upper and lower height of ash layer is presented as grey dots. The lidar retrieved centre of mass for ash is plotted as black dots. For mixed layers where ash is identified with continental aerosol, the height of the layer is presented as light pink dots, and centre of mass are red dots. The height of the planetary boundary layer is shown in violet. Due to weather conditions and technical difficulties the lidar measurements are not a continuous series.**





**Figure 8; Modelled and observed centre of mass for ash at the lidar stations. Green and blue dots represent centre of ash mass, computed from the entire model column, for simulations with and without gravitational settling, shown where ash concentrations were larger than 0.1 μg m⁻³. Magenta and orange represents model centre of ash, calculated above the observed planetary boundary layer (PBL) with and without gravitational settling, respectively. Black dots are corresponding lidar retrieved centre of mass for ash above the PBL (same as Figure 7). Light blue line above indicates where observations are missing.**





**Figure 9: Scatterplots for observed versus simulated centre of ash mass with (magenta) and without gravitational settling (orange). Data correspond to Fig. 8 using model and observed values under 8 km but above the PBL. Correlation between observed and model values is given in the upper left corner.**

