# Peer review of "The operational eEMEP model version 10.4 for volcanic $SO_2$ and ash forecasting"

_Geoscientific Model Development, 2016_

## Short Comment (SC1) · 16 Jan 2017

Dear authors,

in my role as Executive editor of GMD, I would like to bring to your attention our Editorial version 1.1:

http://www.geosci-model-dev.net/8/3487/2015/gmd-8-3487-2015.html

This highlights some requirements of papers published in GMD, which is also available on the GMD website in the 'Manuscript Types' section:

http://www.geoscientific-model-development.net/submission/manuscript_types.html

In particular, please note that for your paper, the following requirement has not been met in the Discussions paper:

- "The main paper must give the model name and version number (or other unique identifier) in the title."

Please add a version number for eEMEP in the title upon your revised submission to GMD.

Yours,

Astrid Kerkweg

---

## Referee Comment (RC1) · Anonymous Referee #1 · 18 Mar 2017

General Comments:

In general, this manuscript is a dense read, mainly because it presents a huge scope of work. Two new species are added to the EMEP MSC-W model, SO2 and volcanic ash, each with their own physics (e.g. sedimentation). Operational use of the model is described in which satellite observations are used to invert for a source term. Sensitivity of the model to meteorological conditions is presented using ensemble forecasts. Finally, results are compared with satellite observations, ground-based lidar, and output from models using other numerical schemes.

This work could be presented as two manuscripts, possibly separating SO2 from ash. But as it is, this manuscript is reasonably well-written, presents material of interest to the journal audience, is sufficiently novel to warrant publication, and passes all the

other limiting criteria (adequate citations, figures of high-quality, sufficient details provided for reproducibility). I recommend that this manuscript be published, but suggest the authors consider a few of the specific comments listed below.

Specific Comments:

At the end of section 2, where the operational set-up is discussed, the authors give a nice description of the scheme used for real-time comparison with satellite retrievals and for inverting for source terms. Section 3 is more of an investigation on the sensitivity of the model to variations in the meteorological conditions by running eEMEP with 24 ensemble cases. At the end of section 3, however, there is a discussion of ensemble runs in an emergency forecast environment. It is not really clear how the ensembles are used in forecasting. Are they available in real-time? It looked like this was just a retrospective on the Bardarbunga event. If ensembles are used, is the source-receptor inversion used with each realization of the ensemble?

It is not really clear what is the range of variability in the ensembles. They seem to be primarily subdivided based on the description of cloud physics. As opposed to explosive eruptions that simultaneously release SO2 and ash, Bardarbunga was primarily fire-fountaining with a continuous surface emission of SO2. I would think that the main discriminating aspect of the meteorology is the characterization of the planetary boundary layer and how vertical diffusivities are calculated. In Hawaii, low-level winds within the boundary layer play a critical role in the SO2 advection, especially the diurnal variations (sea-breeze, nocturnal katabatic winds, etc.). The VMAP project has found that they need to calculate meteorology with WRF at a resolution of 1 km over the Big Island in order to capture the surface winds properly. Is this not as important in Iceland?

Section 4 focuses a bit more than necessary on the benefits of including gravitational sedimentation. It is widely recognized in the volcanic ash dispersion modelling community that it is the dominant removal process for ash > 64 um. It become less and

less important with smaller and smaller particles, to the point where it is negligible compared to the effects of wet scavenging or aggregation. The vertical position of distal ash will be very sensitive to the characterization of the grainsize distribution and on the specific source terms used (mass-loading as a function of height and grainsize at the vent). It is difficult to compare model results with lidar data as evidence supporting including or neglecting sedimentation since the airborne grainsize distribution above the lidar station is not really known.

I would like to commend the authors on making the source code for their model publicly available.

---

## Short Comment (SC2) · 21 Mar 2017

The presentation of the current state of the eEMEP model and its benefit to describe the dispersion of volcanic ash is certainly quite interesting. In particular the attempt to validate an operational model by means of lidar measurements is appreciated – the Eyjafjallajökull eruption is a very good opportunity as the evolution of the ash layer was very well documented. Consequently, there are already numerous papers on this event, some of which pursues similar scientific objectives as Steensen et al.'s manuscript (henceforward referred to as "this paper") i.e. the comparison of model results and observations.

With this short comment I want to suggest to better emphasizing the previous work on this topic. It can be acknowledged in the introduction and in section 4; the latter can

easily be extended to avoid the impression that studies beyond the "Norwegian ash project" (page 11, line 4) are more or less lacking.

Modelling studies related to lidar observations

In excess of the two references in this paper (page 2, lines 30ff) further applications of Eulerian models, which were compared to lidar measurements, include:

- WRF-Chem was used by Webley et al. (2012). Results were e.g. compared to different ash related parameters derived from spaceborne platforms like SEVIRI and Calipso and ground based lidar measurements at Maisach.
- MCCM was used by Emeis et al. (2011) to investigate the transport of ash during the first few days after the eruption. Validation was provided by measurements of the German ceilometer network – a unique opportunity due to its outstanding spatial coverage and resolution.
- COSMO-ART was used by Vogel et al. (2014). They compared the particle number density with in-situ measurements at Hohenpeißenberg and made a qualitative consistency check with lidar measurements at Maisach (Munich).

**Lidar measurements**

Wiegner et al. (2012) described the ash distribution for the same episode covered by this paper. The potential of ground based lidar- and ceilometer measurements was demonstrated and a very good agreement with measurements in Munich and Maisach was found. The comparisons were based on mass concentration profiles. The procedure to estimate mass concentration from lidar backscatter profiles was explained, revealing a complicated inversion because the conversion factors (backscatter  $\rightarrow$  extinction  $\rightarrow$  mass concentration) depend on the microphysics of the particles. Nevertheless we had chosen this way to provide a direct and quantitative intercomparison
including estimates of the uncertainty. Our measurements at Maisach ("Munich") are also considered in this paper. An extension to a larger spatial area was performed in the above mentioned companion paper by Emeis et al., (2011).

**Sidenote**

The EARLINET site "Munich" is in fact "Maisach" (25 km north west of Munich). It is operated by the Ludwig-Maximilians-Universität, Munich; this may be the reason that it is often labeled as "Munich".

**References**

- Emeis, S., Forkel, R., Junkermann, W., Schäfer, K., Flentje, H., Gilge, S., Fricke, W., Wiegner, M., Freudenthaler, V., Groß, S., Ries, L., Meinhardt, F., Birmili, W., Münkel, C., Obleitner, F., and Suppan, P.: Measurement and simulation of the 16/17 April 2010 Eyjafjallajökull volcanic ash layer dispersion in the northern Alpine region, Atmos. Chem. Phys., 11, 2689-2701, doi:10.5194/acp-11-2689-2011, 2011.
- Vogel, H., Förstner, J., Vogel, B., Hanisch, T., Mühr, B., Schättler, U., and Schad, T.: Time-lagged ensemble simulations of the dispersion of the Eyjafjallajökull plume over Europe with COSMO-ART, Atmos. Chem. Phys., 14, 7837-7845, doi:10.5194/acp-14-7837-2014, 2014.
- Webley, P. W., Steensen, T., Stuefer, M., Grell, G., Freitas, S., and Pavolonis, M.: Analyzing the Eyjafjallajökull 2010 eruption using satellite remote sensing, lidar and WRF-Chem dispersion and tracking model, J. Geophys. Res., 117, D00U26, doi:10.1029/2011JD016817, 2012.
- Wiegner, M., J. Gasteiger, S. Groß, F. Schnell, V. Freudenthaler, and R. Forkel: Characterization of the Eyjafjallajökull ash-plume: Potential of lidar
remote sensing, Physics and Chemistry of the Earth 45–46, 79–86, doi: 10.1016/j.pce.2011.01.006, 2012.

---

## Referee Comment (RC2) · Anonymous Referee #2 · 27 Mar 2017

General comments

This manuscript describes the eEMEP model, a version of the EMEP MSC-W CTP, dedicated to emergencies. eEMEP aims at providing rapidly the evolution of a volcanic plume, which can be gaseous (SO2) or particulate (ashes). After a presentation of the operational configuration, The work is divided in two parts. The first one focuses on finding 1) the better way to use ensembles of weather forecasts and 2) a compromise between numerical efficiency and information added by a higher resolution, through the simulation of SO2. The second part focuses on the representation of volcanic ash and in particular on the evaluation of the importance of gravitational sedimentation, and concludes that the sedimentation is finally not so important. In general, this manuscript is very interesting, clear and informative. To me, it can be published as long as the authors address the following questions/remarks.

[Figure]

Specific comments

Section 2.2

(eEMEP is run with 40 or 42 levels. Please precise the corresponding top altitude (even if it is specified in section 4).

Section 3.1

- Is the eruption column between 1500m and 3000m uniform or is there a specific shape ? Does it correspond to what would be done in real time (during an emergency) or is it meant to be as near to the reality as possible ?

- Please define the SO2 'free state' used as initial.

- The sentence about the simple reduction of the meteorological input data is not clear for me. How is the 'representative point (every fourth one) chosen ?

Section 3.2

- Figure 3 is interesting, but it would be very helpfull for the reader to have another one showing the different trajectories according to the different members of the ensembles.

- The authors mention that they believe that a part of the observed SO2 plume is not seen by the model because the emission is older than the beginnning of the run. Maybe. But it would be very easy to prove it by a run beginning 24 hours earlier.

- I fully agree with the conclusions on the compromise to find, to launch ensembles only when the weather is unstable etc. But I think this conclusion is too general. All this work (which is huge!) considers only one meterological situation, one eruption. Maybe the 20x20km is the optimal choice here, but one can not be sure that it will be true under other conditions.

Section 4.1

- please precise how the ash is distributed over the nine bins, to help the reader understanding how the sedimentation will impact fields.

Section 4.3

- In this section, I feel that the authors are more confident in their model than in the observations ! (p 12 line 13 and line 26). I understand they can have some doubts, but I think they should 1) reformulate and 2) ask the people in charge of the observations their expert opinion on the eventual uncertainty of these observations. - It would help to have a (global) idea of the computed gravitational velocity according to the bins. Moreover, the whole study is focused on the position of the ash layer. But does sedimentation impact on the quantity of ash ?

Typo p5, line 23 : Apart form → Apart from

––––––––––––––––––––––––––––––

---

## Author Comment (AC3) · 10 Apr 2017

Response to short comment posted by M. Wiegner

We thank M. Wiegner for taking the time to read and we appreciate the helpful comment and suggestions for improving the manuscript given in this short comment.

Comments are repeated in black, and answers are given in blue.

With this short comment I want to suggest to better emphasizing the previous work on this topic. It can be acknowledged in the introduction and in section 4; the latter can easily be extended to avoid the impression that studies beyond the "Norwegian ash project" (page 11, line 4) are more or less lacking.

The authors agree that including previous work on model comparison to lidar data would be beneficiary for the manuscript. References are added in the text were they are appropriate under the lidar section:
p.12 l.5:
"Webley et al. (2012) found by studying model results from WRF-Chem that ash particles larger than 62.5 µm were not transported further than 120 km from the volcano, indicating that ash particles larger than what are included in this study already have fallen out by the time the air mass reaches the lidar sites and will not affect the observed ash layer. "
p.12 l.13:
"Even though a lidar does not measure concentrations, it is possible to retrieve these using mass-to-extinction coefficients. Ansmann et al. (2011) and Wiegner et al. (2012) estimated maximum ash concentrations of around 1100 $\mu gm^{-3}$ with around 40 % uncertainty over Hamburg and Munich (lidar situated actually at Maisach) on 17 April respectively, at similar times when maximum concentrations where found in our model results."
p.12 l. 15
"The model shows this shift in ash height from the higher first part of the plume to the lower second part of the plume for all the stations, and this is also found in several other ash transport model comparisons to lidar observations over Europe (Emeis et al., 2011; Folch et al., 2012; Webley et al., 2012; Vogel et al., 2014)."

Sidenote

The EARLINET site "Munich" is in fact "Maisach" (25 km north west of Munich). It is operated by the Ludwig-Maximilians-Universität, Munich; this may be the reason that it is often labeled as "Munich".

The authors are thankful for pointing this out, and further explanation for this. Since in the dataset the station is labelled Munich, this is the name that is used in the manuscript, but with an explanation that the station is actually situated in Maisach (see above).

References:

Ansmann, A., Tesche, M., Seifert, P., Gross, S., Freudenthaler, V., Apituley, A., ... & Hiebsch, A.: Ash and fine-mode particle mass profiles from EARLINET-AERONET observations over central Europe after the eruptions of the Eyjafjallajökull volcano in 2010. Journal of Geophysical Research: Atmospheres, 116(D20), 2011.

Emeis, S., Forkel, R., Junkermann, W., Schäfer, K., Flentje, H., Gilge, S., Fricke, W., Wiegner, M., Freudenthaler, V., Groß, S., Ries, L., Meinhardt, F., Birmili, W., Münkel, C., Obleitner, F., and Suppan, P.: Measurement and simulation of the 16/17 April 2010 Eyjafjallajökull volcanic ash layer dispersion in the northern Alpine region, Atmos. Chem. Phys., 11, 2689-2701, doi:10.5194/acp-11-2689- 2011, 2011.

Folch, A., Costa, A., & Basart, S: Validation of the FALL3D ash dispersion model using observations of the 2010 Eyjafjallajökull volcanic ash clouds. *Atmospheric Environment*, *48*, 165-183, 2012.

Vogel, H., Förstner, J., Vogel, B., Hanisch, T., Mühr, B., Schättler, U., and Schad, T.: Time-lagged ensemble simulations of the dispersion of the Eyjafjallajökull plume over Europe with COSMO-ART, Atmos. Chem. Phys., 14, 7837-7845, doi:10.5194/acp-14-7837-2014, 2014

Webley, P. W., Steensen, T., Stuefer, M., Grell, G., Freitas, S., and Pavolonis, M.: Analyzing the Eyjafjallajökull 2010 eruption using satellite remote sensing, lidar and WRF-Chem dispersion and tracking model, J. Geophys. Res., 117, D00U26, doi:10.1029/2011JD016817, 2012.

Wiegner, M., J. Gasteiger, S. Groß, F. Schnell, V. Freudenthaler, and R. Forkel: Characterization of the Eyjafjallajökull ash-plume: Potential of lidar remote sensing, Physics and Chemistry of the Earth 45–46, 79–86, doi: 10.1016/j.pce.2011.01.006, 2012.

---

## Author Comment (AC4) · 10 Apr 2017

The comment was uploaded in the form of a supplement:
http://www.geosci-model-dev-discuss.net/gmd-2016-315/gmd-2016-315-AC4-supplement.pdf

---

## Author Response (AR1)

Response to Review #1

We thank the reviewer for taking the time to read and we appreciate the helpful comment and suggestions for improving the manuscript given in this review.

Answers to the specific comments are given below, reviewer comments are given in black, answers are given in blue, and changes in the manuscript are noted in quotations (""), also in blue.

Specific Comments:

At the end of section 2, where the operational set-up is discussed, the authors give a nice description of the scheme used for real-time comparison with satellite retrievals and for inverting for source terms. Section 3 is more of an investigation on the sensitivity of the model to variations in the meteorological conditions by running eEMEP with 24 ensemble cases. At the end of section 3, however, there is a discussion of ensemble runs in an emergency forecast environment. It is not really clear how the ensembles are used in forecasting. Are they available in real-time? It looked like this was just a retrospective on the Bardarbunga event. If ensembles are used, is the source-receptor inversion used with each realization of the ensemble?

The authors are grateful for the reviewer pointing out that this is not clear in the text. Running an ensemble forecasts is very computationally demanding, especially on a high resolution. The ensemble meteorology is available in real-time, and dispersion forecasts may therefore be run if the ensemble data show a large spread in the placement of pressures systems that can influence the dispersion of the volcanic emission. For the emission estimate, this is another source of uncertainty that is possibly bigger than that caused by the weather. Running an inversion to produce a new source term for all the ensemble members is probably too computationally demanding, and running the ensemble members with different source term would not reflect the uncertainty due to only the weather as presented in the manuscript. To include this aspect in the description, this sentence will be added on p. 8, l. 4

"Using here the same best guess source term in all our ensemble model simulations offers the opportunity to study the results based on only the different weather situations, meteorology uncertainties and resolution."

And these two sentences will be added in the manuscript p. 10, l. 23.

"As in this study, to exclusively look at the spread due to the uncertainty in the weather forecast, the same source term should be used in all the members. Therefore the model simulations used as input for the inversion calculations will only be driven by the deterministic meteorology."

It is not really clear what is the range of variability in the ensembles. They seem to be primarily subdivided based on the description of cloud physics. As opposed to explosive eruptions that simultaneously release SO2 and ash, Bardarbunga was primarily fire-fountaining with a continuous surface emission of SO2. I would think that the main discriminating aspect of the meteorology is the characterization of the planetary boundary layer and how vertical diffusivities are calculated. In Hawaii, lowlevel winds within the boundary layer play a critical role in the SO2 advection, especially the diurnal variations (sea-breeze, nocturnal katabatic winds, etc.). The VMAP project has found that they need to calculate meteorology with WRF at a resolution of 1 km over the Big Island in order to capture the surface winds properly. Is this not as important in Iceland?

Of the 24 ensemble members used in the study, they are divided over two cloud physics parameterization, these are again divided over two forecast start points were six of each model parameterization start at 00 UTC and 12 UTC and the last six of each start at 06 UTC and 18 UTC. The six remaining members with equal model parameterization and start time are perturbed using the EuroTEPS (European Targeted Ensemble Prediction system) that perturbed the members both in the initial field and on the model domain border (Frogner and Iversen, 2010). We hope that will be made clearer in the manuscript by adding on p. 8, l. 20:

"All of the members are perturbed by using EuroTEPS."

Comparing the VMAP project results (Businger et al., 2015) to this study, the strength of the Barðarbunga eruption, especially at the start of the eruption period is stronger releasing more $SO_2$ higher up in the atmosphere (700 m for VMAP and up to 3000 m assumed here). The area of interest is also different, where the VMAP hopes to forecast the pollution due to the volcanic degassing on a local scale, the scope of this study is to investigate the regional scale transportation of volcanic emission. Gíslason et al. (2015) investigates the environmental pressure on Iceland due to the Barðarbunga eruption with a 4 km model resolution, and found that only a fraction of the released $SO_2$ up to 500 m (10-20% of total) is affecting the surface concentration at distances up to hundreds of km from the volcano. The geographical location of Iceland also makes the dispersion more influenced by large scale pressure systems compared to the meteorological conditions Hawaii experiences. During the Barðarbunga eruption most of the pollution was transported far away from the volcano.

Section 4 focuses a bit more than necessary on the benefits of including gravitational sedimentation. It is widely recognized in the volcanic ash dispersion modelling community that it is the dominant removal process for ash > 64 um. It become less and less important with smaller and smaller particles, to the point where it is negligible compared to the effects of wet scavenging or aggregation. The vertical position of distal ash will be very sensitive to the characterization of the grainsize distribution and on the specific source terms used (mass-loading as a function of height and grainsize at the vent). It is difficult to compare model results with lidar data as evidence supporting including or neglecting sedimentation since the airborne grainsize distribution above the lidar station is not really known.

The authors agree that it is difficult to validate sedimentation inclusion. However lidar measurements are the only ones which offer a chance to detect vertical displacements of ash plumes, something which should happen as a consequence of significant sedimentation. We believe its an original idea, and do not claim, that it's the ultimate idea to validate sedimentation. The study shows that subsidence due to a high pressure is much more important than the vertical displacement caused by gravitational settling. The authors still find it necessary to include the process of gravitational settling for a more correct description of the ash transport. A 1 km displacement as found in our study is important if the vertical wind shear is strong. Wet scavenging is already included in the model, and wet deposition of e.g. $SO_x$ is validated every year in the EMEP MSC-W reports and are therefore not included in this paper. Aggregation of fine ash to large ash is still not included in the model.

The manuscript has been extended in this section to include other model comparisons to lidar measurements during the Eyjafjallajökull eruption as suggested by the short comment 2. One of them were from Webley et al. (2012), where WRF-Chem simulations show that ash particles larger than 62.5 μm were not transported further than 120 km from the volcano. This may indicate that including larger ash particles would be extensive in model simulations were the goal is to asses the airspace over Norway.

References:

Businger, S., Huff, R., Pattantyus, A., Horton, K., Sutton, A. J., Elias, T., & Cherubini, T.: Observing and Forecasting Vog Dispersion from Kīlauea Volcano, Hawaii. Bulletin of the American Meteorological Society, 96(10), 1667-1686, 2015.

Frogner, I-L., and Iversen, T. EuroTEPS—a targeted version of ECMWF EPS for the European area. Tellus A 63.3 415-428, 2011.

Gíslason, S. R., Stefánsdóttir, G., Pfeffer, M. A., Barsotti, S.,Jóhannsson, T., Galeczka, I., Bali, E., Sigmarsson, O., Stefánsson, A., Keller, N. S., Sigurdsson, Á., Bergsson, B., Galle, B., Jacobo, V. C., Arellano, S., Aiuppa, A., Jónasdóttir, E. B., Eiríksdóttir, E. S., Jakobsson, S., Guðfinnsson, G. H., Halldórsson, S. A., Gunnarsson, H., Haddadi, B., Jónsdóttir, I., Thordarson, T., Riishuus, M., Högnadóttir, T., Dürig, T., Pedersen, G. B. M., Höskuldsson, Á., and Gudmundsson, M. T.: Environmental pressure from the 2014–15 eruption of Bárðarbunga volcano, Iceland, Geochem. Persp. Lett., 1, 84–93, 2015.

Webley, P. W., Steensen, T., Stuefer, M., Grell, G., Freitas, S., and Pavolonis, M.: Analyzing the Eyjafjallajökull 2010 eruption using satellite remote sensing, lidar and WRF-Chem dispersion and tracking model, J. Geophys. Res., 117, D00U26, doi:10.1029/2011JD016817, 2012.

Response to Reviewer #2

We thank the reviewer for taking the time to read our manuscript and we appreciate the helpful comment and suggestions for improving the manuscript given in this review.

Answers to the specific comments are given below, reviewer comments are given in black, answers are given in blue, and changes in the manuscript are noted in quotations (""), also in blue.

Specific Comments:

Section 2.2

eEMEP is run with 40 or 42 levels. Please precise the corresponding top altitude (even if it is specified in section 4).

The authors agree that this is worth noting, and will add the corresponding heights and change p.5 l.3 to:

"Model simulations presented in this paper are either done with 40 or 42 vertical levels depending on available meteorology pre-processing, with a model top at 32 km and 30 km respectively."

Section 3.1

Is the eruption column between 1500m and 3000m uniform or is there a specific shape? Does it correspond to what would be done in real time (during an emergency) or is it meant to be as near to the reality as possible?

The emission flux is distributed uniformly over the eruption column. During an emergency, either the source term is from inversion calculations and then may include specification of the height of emissions or an a priori height distribution estimate will be used. However, here we did not vary nor test the a priori emission height, and rather wanted to be near to reality. The sentences changed in the manuscript:

"…emission estimate with a 120 kt d-1 flux uniformly over an eruption column between 1500 m to 3000 m matched best for the first days of September. This emission term is also supported by Thordarson and Hartley (2015) and used here. In an emergency case an a priori source term would be used first when little information about the volcanic source term is known. Using here the same best guess source term in all our ensemble model simulations offers the opportunity to study the results based on only the different weather situations, meteorology uncertainties and resolution. "

Please define the SO2 'free state' used as initial.

Free state means that there is no volcanic $SO_2$ in the atmosphere at the start of the forecast. This will be added to p.8 l. 29:

"In contrast to what is possibly done for a real case, all the forecasts are started from a model state with no volcanic $SO_2$ in the atmosphere,…"

The sentence about the simple reduction of the meteorological input data is not clear for me. How is the 'representative point (every fourth one) chosen?

This is a very simple reduction where the algorithm loops in both horizontal directions trough the original grid and picks out every second and every fourth grid point to obtain the values of the grid points in the coarser grid resolutions,. As stated in the paper, this may not provide a smooth field and some maximum/minimum values may be representing a larger area than originally found by the NWP. Sentence changed to:

"A simple reduction in resolution of the meteorological input data is obtained here by letting every other or every fourth original grid value become the grid value representing the coarser grid resolution respectively."

Section 3.2

Figure 3 is interesting, but it would be very helpful for the reader to have another one showing the different trajectories according to the different members of the ensembles.

Since eEMEP is an Eulerian model it is not possible to compute single trajectories, however the authors agree that including a figure showing volcanic $SO_2$ VCDs from different ensemble members would increase the understanding of the results. A new figure has been created and will be added as figure 4 in the manuscript, showing the 5 DU contour line for four of the members, one from each of the model parameterizations and starting times (figure introduced here as fig 1, see below).

[Figure]

**Figure 1: 5 DU contour lines for four exemplary members after 48 hours of forecast in the low, mid and high resolution ensembles, in the left, middle and right column respectively, for start time 00 UTC 3. September (panels a,b,c), 00 UTC 4. September (panels d,e,f) and 00 UTC 5 September (panels g,h,i).**

This Figure will be added to the manuscript in addition to additions to the paragraph starting on p.9 .23:

"The difference in the spread is also seen to be weather dependent especially when using a low threshold. Figure 4 shows the 5 DU contour line for the forecasts corresponding to Figure 2, for four of the ensemble members. Each of the four members represents one of the perturbed members from the two different model parameterisations and starting times. For the first forecast started there are large differences between the members for areas where they have VCDs above 5 DU. In the second forecast started, the differences between the members are smaller, while the last forecast from 5 Sept 00 UTC shows that, although the members all have plumes with VCDs over 5 DU going south from Iceland, they have quite different positions indicating a different position of the low pressure system. "

The authors mention that they believe that a part of the observed SO2 plume is not seen by the model because the emission is older than the beginning of the run. Maybe. But it would be very easy to prove it by a run beginning 24 hours earlier.

This has been proven before. The evolution of this plume has been studied in Schmidt et al. (2016) where both satellite and surface concentrations are compared to NAME model results for September 5. Steensen et al. (2016) also studies the evolution of the plume over the three first months of the fissure eruption. The authors agree that the manuscript is not clear from the manuscript and will add the previous studies on the evolutions. P 10 l. 12 will be changed to:

"An area with high $SO_2$ concentrations in the southwest is not captured by either of the forecasts. Previous studies of this eruption (Schmidt et al., 2015, Steensen et al., 2016) show that this area is affected by older emissions compared to what is included in our model simulations that start 00 UTC 4 September, and is thus not apparent in the model simulations presented here."

I fully agree with the conclusions on the compromise to find, to launch ensembles only when the weather is unstable etc. But I think this conclusion is too general. All this work (which is huge!) considers only one meteorological situation, one eruption. Maybe the 20x20km is the optimal choice here, but one can not be sure that it will be true under other conditions.

The authors agree that to fully conclude on this, more meteorological situations and eruption styles should be studied than what was feasible in this study and will make the statement less general, and add at the end of the paragraph:

"Ideally more studies should be done, that include other weather situations as well as different types of eruptions to conclude on the best grid resolution."

Section 4.1

please precise how the ash is distributed over the nine bins, to help the reader understanding how the sedimentation will impact fields.

The size distribution is:

4 µm 16 %, 6 µm 18 %, 8 µm 15 %, 10 µm 13 %, 12 µm 10 %, 14 µm 8%, 16 µm 6 %, 18 µm 7 %, 25 µm 7 %.

, and will be added to the manuscript p.10 l.

"The ash is distributed over nine size bins with characteristic size of 4, 6, 8, 10, 12, 14, 16, 18 and 25 µm and the ash in the source term is distributed among the bins as follows: 16, 18, 15, 13, 10, 8, 6, 7 and 7%".

Section 4.3

In this section, I feel that the authors are more confident in their model than in the observations! (p 12 line 13 and line 26). I understand they can have some doubts, but I think they should 1) reformulate and 2) ask the people in charge of the observations their expert opinion on the eventual uncertainty of these observations. - It

would help to have a (global) idea of the computed gravitational velocity according to the bins. Moreover, the whole study is focused on the position of the ash layer. But does sedimentation impact on the quantity of ash?

Concerning the measurements at Cabauw: Figure 5 in Pappalardo et al. (2013) shows that there were no measurements taken during this time, and the study also commented on that low clouds often prevented observations at this station. The sentence p.12 l. 13 is changed in the manuscript:

"At Cabauw, the first part of the ash plume is not covered by the lidar because no measurements are available, while the second part shows similar simulated and observed level of maximum concentrations."

For the Leipzig stations, no reason is given for the apparent outliers in Pappalardo et al. (2013), however the centres of mass oscillates from 4 km to 12 km in an unphysical way, and centres of mass or even the height of the ash layer are below 12 km for all the other stations. The sentence p.12 l. 26 is changed in the manuscript:

"In Leipzig a few observations of centre of mass on 16 and 18 April are much higher (at 12 km) than the model centre of mass heights and the corresponding heights at the other stations at this time, implying these high altitude measurements may not represent ash."

Figure 2 shows the ash concentrations for the model simulation with no gravitational settling, equivalent to Figure 7 in the manuscript. The difference in quantity of ash between the two model simulations with and without gravitational settling is minimal, which is supported by the small difference in centre of ash layer. This is because the size range only includes the very small sizes of ash, up to 25 μm. Model comparison studies where also the coarse ash (2 mm > d > 64 μm ) is included in the model simulations show that these larger ash particles fall out before reaching central Europe and the lidar stations (Webley et al., 2012). This is in agreement with the relatively minor fall speed for fine ash particles (d < 64 μm) around 0.01 km/h (Bonadonna et al., 1998; Rose et al., 2001)

[Figure]

**Figure 2: Height-time profiles of ash concentrations from eEMEP model, not including gravitational settling, at the six EARLINET lidar stations in April-May 2010 episode (contour graph in background). Lidar-detected upper and lower height of ash layer is presented as grey dots. The lidar retrieved centre of mass for ash is plotted as black dots. For mixed layers where ash is identified with continental aerosol, the height of the layer is presented as light pink dots, and centre of mass are red dots. The height of the planetary boundary layer is shown in violet. Due to weather conditions and technical difficulties the lidar measurements are not a continuous series.**

Typo p5, line 23 : Apart form → Apart from

Changed accordingly.

Please add a version number for eEMEP in the title upon your revised submission to GMD.

The version number has been added to the title and the new title is now:

"The operational eEMEP model version 10.4 for volcanic $SO_2$ and ash forecasting"

Response to short comment posted by M. Wiegner

We thank M. Wiegner for taking the time to read and we appreciate the helpful comment and suggestions for improving the manuscript given in this short comment.

Comments are repeated in black, and answers are given in blue.

With this short comment I want to suggest to better emphasizing the previous work on this topic. It can be acknowledged in the introduction and in section 4; the latter can easily be extended to avoid the impression that studies beyond the "Norwegian ash project" (page 11, line 4) are more or less lacking.

The authors agree that including previous work on model comparison to lidar data would be beneficiary for the manuscript. References are added in the text were they are appropriate under the lidar section:
p.12 l.5:
"Webley et al. (2012) found by studying model results from WRF-Chem that ash particles larger than 62.5 µm were not transported further than 120 km from the volcano, indicating that ash particles larger than what are included in this study already have fallen out by the time the air mass reaches the lidar sites and will not affect the observed ash layer. "
p.12 l.13:
"Even though a lidar does not measure concentrations, it is possible to retrieve these using mass-to-extinction coefficients. Ansmann et al. (2011) and Wiegner et al. (2012) estimated maximum ash concentrations of around 1100 $\mu gm^{-3}$ with around 40 % uncertainty over Hamburg and Munich (lidar situated actually at Maisach) on 17 April respectively, at similar times when maximum concentrations where found in our model results."
p.12 l. 15
"The model shows this shift in ash height from the higher first part of the plume to the lower second part of the plume for all the stations, and this is also found in several other ash transport model comparisons to lidar observations over Europe (Emeis et al., 2011; Folch et al., 2012; Webley et al., 2012; Vogel et al., 2014)."

Sidenote

The EARLINET site "Munich" is in fact "Maisach" (25 km north west of Munich). It is operated by the Ludwig-Maximilians-Universität, Munich; this may be the reason that it is often labeled as "Munich".

The authors are thankful for pointing this out, and further explanation for this. Since in the dataset the station is labelled Munich, this is the name that is used in the manuscript, but with an explanation that the station is actually situated in Maisach (see above).

[revised manuscript text omitted]

**OMI satellite retrieval 5 Sept**   **low 8-16 UTC 5 Sept**   **high 8-16 UTC 5 Sept**

DU
frequency over 10 DU
frequency over 10 DU

0.1 0.5 1 1.5 2 3 5 10
2 4 6 8 10 12 14 16 18 20 22 24
2 4 6 8 10 12 14 16 18 20 22 24

[Figure]

**Figure 4: 5 DU contour lines for four exemplary members after 48 hours of forecast in the low, mid and high resolution ensembles, in the left, middle and right column respectively, for start time 00 UTC 3. September (panels a,b,c), 00 UTC 4. September (panels d,e,f) and 00 UTC 5 September (panels g,h,i).**

[Figure]

**Figure 5:** OMI retrieval of $SO_2$ (left) for the satellite overpasses between 8 UTC and 16 UTC 5 Sept. Frequency of ensemble members over 10 DU for the low (middle) and high (right) resolution runs. Frequencies are computed every hour and averaged over the same time period, using the forecast runs started 4 September 00 UTC.

[Figure]

Figure 56: Mean ash column burdens from 8 to 9 UTC 16 April for SEVIRI and IASI satellite ash retrievals, and eEMEP, SNAP and FLEXPART model simulations.

[Figure]

**Figure 6̶7: Map of EARLINET lidar measurement sites used in the study.**

[Figure]

**Figure 7,8:** Height-time profiles of ash concentrations from eEMEP model, including gravitational settling, at the six EARLINET lidar stations (see figure 6) in April-May 2010 episode (contour graph in background). Lidar-detected upper and lower height of ash layer is presented as grey dots. The lidar retrieved centre of mass for ash is plotted as black dots. For mixed layers where ash is identified with continental aerosol, the height of the layer is presented as light pink dots, and centre of mass are red dots. The height of the planetary boundary layer is shown in violet. Due to weather conditions and technical difficulties the lidar measurements are not a continuous series.

[Figure]

**Figure 8,9: Modelled and observed centre of mass for ash at the lidar stations. Green and blue dots represent centre of ash mass, computed from the entire model column, for simulations with and without gravitational settling, shown where ash concentrations were larger than 0.1 $\mu$g m$^{-3}$. Magenta and orange represents model centre of ash, calculated above the observed planetary boundary layer (PBL) with and without gravitational settling, respectively. Black dots are corresponding lidar retrieved centre of mass for ash above the PBL (same as Figure 7). Light blue line above indicates where observations are missing.**

[Figure]

**Figure 910:** Scatterplots for observed versus simulated centre of ash mass with (magenta) and without gravitational settling (orange). Data correspond to Fig. 8 using model and observed values under 8 km but above the PBL. Correlation between observed and model values is given in the upper left corner.